# Content-based recommendations
# with Poisson factorization

**Prem Gopalan**
Department of Computer Science
Princeton University
Princeton, NJ 08540
pgopalan@cs.princeton.edu

**Laurent Charlin**
Department of Computer Science
Columbia University
New York, NY 10027
lcharlin@cs.columbia.edu

**David M. Blei**
Departments of Statistics & Computer Science
Columbia University
New York, NY 10027
david.blei@columbia.edu

## Abstract

We develop collaborative topic Poisson factorization (CTPF), a generative model
of articles and reader preferences. CTPF can be used to build recommender sys-
tems by learning from reader histories and content to recommend personalized
articles of interest. In detail, CTPF models both reader behavior and article texts
with Poisson distributions, connecting the latent topics that represent the texts
with the latent preferences that represent the readers. This provides better recom-
mendations than competing methods and gives an interpretable latent space for
understanding patterns of readership. Further, we exploit stochastic variational
inference to model massive real-world datasets. For example, we can fit CPTF
to the full arXiv usage dataset, which contains over 43 million ratings and 42
million word counts, within a day. We demonstrate empirically that our model
outperforms several baselines, including the previous state-of-the art approach.

## 1   Introduction

In this paper we develop a probabilistic model of articles and reader behavior data. Our model is
called *collaborative topic Poisson factorization* (CTPF). It identifies the latent topics that under-
lie the articles, represents readers in terms of their preferences for those topics, and captures how
documents about one topic might be interesting to the enthusiasts of another.

As a recommendation system, CTPF performs well in the face of massive, sparse, and long-tailed
data. Such data is typical because most readers read or rate only a few articles, while a few readers
may read thousands of articles. Further, CTPF provides a natural mechanism to solve the "cold start"
problem, the problem of recommending previously unread articles to existing readers. Finally, CTPF
provides a new exploratory window into the structure of the collection. It organizes the articles
according to their topics and identifies important articles both in terms of those important to their
topic and those that have transcended disciplinary boundaries.

We illustrate the model with an example. Consider the classic paper "Maximum likelihood from
incomplete data via the EM algorithm" [5]. This paper, published in the Journal of the Royal Sta-
tistical Society (B) in 1977, introduced the expectation-maximization (EM) algorithm. The EM
algorithm is a general method for finding maximum likelihood estimates in models with hidden
random variables. As many readers will know, EM has had an enormous impact on many fields,

including computer vision, natural language processing, and machine learning. This original paper has been cited over 37,000 times.

Figure 1 illustrates the CTPF representation of the EM paper. (This model was fit to the shared libraries of scientists on the Mendeley website; the number of readers is 80,000 and the number of articles is 261,000.) In the figure, the horizontal axes contains topics, latent themes that pervade the collection [2]. Consider the black bars in the left figure. These represent the topics that the EM paper is about. (These were inferred from the abstract of the paper.) Specifically, it is about probabilistic modeling and statistical algorithms. Now consider the red bars on the right, which are summed with the black bars. These represent the preferences of the readers who have the EM paper in their libraries. CTPF has uncovered the interdisciplinary impact of the EM paper. It is popular with readers interested in many fields outside of those the paper discusses, including computer vision and statistical network analysis.

The CTPF representation has advantages. For forming recommendations, it naturally interpolates between using the text of the article (the black bars) and the inferred representation from user behavior data (the red bars). On one extreme, it recommends rarely or never read articles based mainly on their text; this is the cold start problem. On the other extreme, it recommends widely-read articles based mainly on their readership. In this setting, it can make good inferences about the red bars. Further, in contrast to traditional matrix factorization algorithms, we combine the space of preferences and articles via interpretable topics. CTPF thus offers reasons for making recommendations, readable descriptions of reader preferences, and an interpretable organization of the collection. For example, CTPF can recognize the EM paper is among the most important statistics papers that has had an interdisciplinary impact.

In more detail, CTPF draws on ideas from two existing models: collaborative topic regression [20] and Poisson factorization [9]. Poisson factorization is a form of probabilistic matrix factorization [17] that replaces the usual Gaussian likelihood and real-valued representations with a Poisson likelihood and non-negative representations. Compared to Gaussian factorization, Poisson factorization enjoys more efficient inference and better handling of sparse data. However, PF is a basic recommendation model. It cannot handle the cold start problem or easily give topic-based representations of readers and articles.

Collaborative topic regression is a model of text and reader data that is based on the same intuitions as we described above. (Wang and Blei [20] also use the EM paper as an example.) However, in its implementation, collaborative topic regression is a non-conjugate model that is complex to fit, difficult to work with on sparse data, and difficult to scale without stochastic optimization. Further, it is based on a Gaussian likelihood of reader behavior. Collaborative Poisson factorization, because it is based on Poisson and gamma variables, enjoys an easier-to-implement and more efficient inference algorithm and a better fit to sparse real-world data. As we show below, it scales more easily and provides significantly better recommendations than collaborative topic regression.

## 2   The collaborative topic Poisson factorization model

In this section we describe the collaborative topic Poisson factorization model (CTPF), and discuss its statistical properties. We are given data about users (readers) and documents (articles), where each user has read or placed in his library a set of documents. The rating $r_{ud}$ equals one if user $u$ consulted document $d$, can be greater than zero if the user rated the document and is zero otherwise. Most of the values of the matrix $y$ are typically zero, due to sparsity of user behavior data.

**Background: Poisson factorization.**   CTPF builds on Poisson matrix factorization [9]. In collaborative filtering, Poisson factorization (PF) is a probabilistic model of users and items. It associates each user with a latent vector of preferences, each item with a latent vector of attributes, and constrains both sets of vectors to be sparse and non-negative. Each cell of the observed matrix is assumed drawn from a Poisson distribution, whose rate is a linear combination of the corresponding user and item attributes. Poisson factorization has also been used as a topic model [3], and developed as an alternative text model to latent Dirichlet allocation (LDA). In both applications Poisson factorization has been shown to outperform competing methods [3, 9]. PF is also more easily applicable to real-life preference datasets than the popular Gaussian matrix factorization [9].

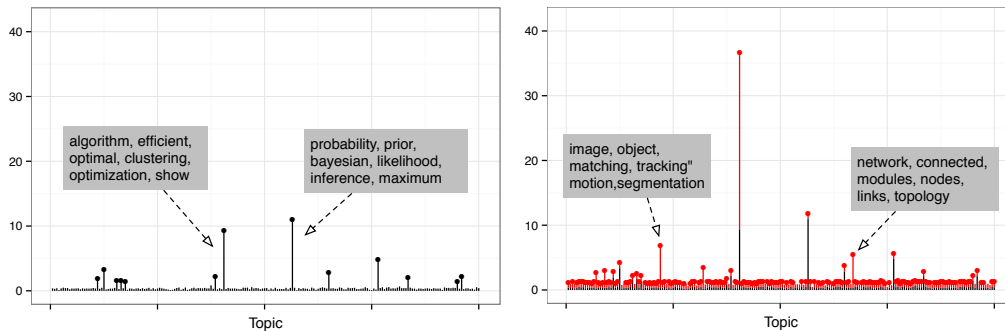

Figure 1: We visualized the inferred topic intensities $\theta$ (the black bars) and the topic offsets $\epsilon$ (the red bars) of an article in the Mendeley [13] dataset. The plots are for the statistics article titled "Maximum likelihood from incomplete data via the EM algorithm". The black bars represent the topics that the EM paper is about. These include probabilistic modeling and statistical algorithms. The red bars represent the preferences of the readers who have the EM paper in their libraries. It is popular with readers interested in many fields outside of those the paper discusses, including computer vision and statistical network analysis.

**Collaborative topic Poisson factorization.** CTPF is a latent variable model of user ratings and document content. CTPF uses Poisson factorization to model both types of data. Rather than modeling them as independent factorization problems, we connect the two latent factorizations using a correction term [20] which we'll describe below.

Suppose we have data containing $D$ documents and $U$ users. CTPF assumes a collection of $K$ unnormalized *topics* $\beta_{1:K}$. Each topic $\beta_k$ is a collection of word intensities on a vocabulary of size $V$. Each component $\beta_{vk}$ of the unnormalized topics is drawn from a Gamma distribution. Given the topics, CTPF assumes that a document $d$ is generated with a vector of $K$ latent *topic intensities* $\theta_d$, and represents users with a vector of $K$ latent *topic preferences* $\eta_u$. Additionally, the model associates each document with $K$ latent *topic offsets* $\epsilon_d$ that capture the document's deviation from the topic intensities. These deviations occur when the content of a document is insufficient to explain its ratings. For example, these variables can capture that a machine learning article is interesting to a biologist, because other biologists read it.

We now define a generative process for the observed word counts in documents and observed user ratings of documents under CTPF:

1. **Document model:**
   (a) Draw topics $\beta_{vk} \sim \text{Gamma}(a, b)$
   (b) Draw document topic intensities $\theta_{dk} \sim \text{Gamma}(c, d)$
   (c) Draw word count $w_{dv} \sim \text{Poisson}(\theta_d^T \beta_v)$.

2. **Recommendation model:**
   (a) Draw user preferences $\eta_{uk} \sim \text{Gamma}(e, f)$
   (b) Draw document topic offsets $\epsilon_{dk} \sim \text{Gamma}(g, h)$
   (c) Draw $r_{ud} \sim \text{Poisson}(\eta_u^T(\theta_d + \epsilon_d))$.

CTPF specifies that the conditional probability that a user $u$ rated document $d$ with rating $r_{ud}$ is drawn from a Poisson distribution with rate parameter $\eta_u^T(\theta_d + \epsilon_d)$. The form of the factorization couples the user preferences for the document topic intensities $\theta_d$ and the document topic offsets $\epsilon_d$. This allows the user preferences to be interpreted as affinity to latent topics.

CTPF has two main advantages over previous work (e.g., [20]), both of which contribute to its superior empirical performance (see Section 5). First, CTPF is a conditionally conjugate model when augmented with auxiliary variables. This allows CTPF to conveniently use standard variational inference with closed-form updates (see Section 3). Second, CTPF is built on Poisson factorization; it can take advantage of the natural sparsity of user consumption of documents and can analyze massive real-world data. This follows from the likelihood of the observed data under the model [9].

We analyze user preferences and document content with CTPF via its posterior distribution over latent variables $p(\beta_{1:K}, \theta_{1:D}, \epsilon_{1:D}, \eta_{1:U} | \boldsymbol{w}, \boldsymbol{r})$. By estimating this distribution over the latent structure, we can characterize user preferences and document readership in many useful ways. Figure 1 gives an example.

**Recommending old and new documents.** Once the posterior is fit, we use CTPF to recommend *in-matrix* documents and *out-matrix* or *cold-start* documents to users. We define in-matrix documents as those that have been rated by at least one user in the recommendation system. All other documents are new to the system. A cold-start recommendation of a new document is based entirely on its content. For predicting both in-matrix and out-matrix documents, we rank each user's unread documents by their posterior expected Poisson parameters,

$$\text{score}_{ud} = \text{E}[\eta_u^T(\theta_d + \epsilon_d) | \boldsymbol{w}, \boldsymbol{r}]. \tag{1}$$

The intuition behind the CTPF posterior is that when there is no reader data, we depend on the topics to make recommendations. When there is both reader data and article content, this gives information about the topic offsets. We emphasize that under CTPF the in-matrix recommendations and cold-start recommendations are not disjoint tasks. There is a continuum between these tasks. For example, the model can provide better predictions for articles with few ratings by leveraging its latent topic intensities $\theta_d$.

## 3 Approximate posterior inference

Given a set of observed document ratings $\boldsymbol{r}$ and their word counts $\boldsymbol{w}$, our goal is to infer the topics $\beta_{1:K}$, the user preferences $\eta_{1:U}$, the document topic intensities $\theta_{1:D}$, the document topic offsets $\epsilon_{1:D}$. With estimates of these quantities, we can recommend in-matrix and out-matrix documents to users.

Computing the exact posterior distribution $p(\beta_{1:K}, \theta_{1:D}, \epsilon_{1:D}, \eta_{1:U} | \boldsymbol{w}, \boldsymbol{r})$ is intractable; we use variational inference [15]. We first develop a coordinate ascent algorithm—a batch algorithm that iterates over only the non-zero document-word counts and the non-zero user-document ratings. We then present a more scalable stochastic variational inference algorithm.

In variational inference we first define a parameterized family of distributions over the hidden variables. We then fit the parameters to find a distribution that minimizes the KL divergence to the posterior. The model is conditionally conjugate if the complete conditional of each latent variable is in the exponential family and is in the same family as its prior. (The complete conditional is the conditional distribution of a latent variable given the observations and the other latent variables in the model [8].) For the class of conditionally conjugate models, we can perform this optimization with a coordinate ascent algorithm and closed form updates.

**Auxiliary variables.** To facilitate inference, we first augment CTPF with auxiliary variables. Following Ref. [6] and Ref. [9], we add $K$ latent variables $z_{dv,k} \sim \text{Poisson}(\theta_{dk}\beta_{vk})$, which are integers such that $w_{dv} = \sum_k z_{dv,k}$. Similarly, for each observed rating $r_{ud}$, we add $K$ latent variables $y_{ud,k}^a \sim \text{Poisson}(\eta_{uk}\theta_{dk})$ and $K$ latent variables $y_{ud,k}^b \sim \text{Poisson}(\eta_{uk}\epsilon_{dk})$ such that $r_{ud} = \sum_k y_{ud,k}^a + y_{ud,k}^b$. A sum of independent Poisson random variables is itself a Poisson with rate equal to the sum of the rates. Thus, these new latent variables preserve the marginal distribution of the observations, $w_{dv}$ and $r_{ud}$. Further, when the observed counts are 0, these auxiliary variables are not random. Consequently, our inference procedure need only consider the auxiliary variables for non-zero observations.

CTPF with the auxiliary variables is conditionally conjugate; its complete conditionals are shown in Table 1. The complete conditionals of the Gamma variables $\beta_{vk}$, $\theta_{dk}$, $\epsilon_{dk}$, and $\eta_{uk}$ are Gamma distributions with shape and rate parameters as shown in Table 1. For the auxiliary Poisson variables, observe that $z_{dv}$ is a $K$-dimensional latent vector of Poisson counts, which when conditioned on their observed sum $w_{dv}$, is distributed as a multinomial [14, 4]. A similar reasoning underlies the conditional for $y_{ud}$ which is a $2K$-dimensional latent vector of Poisson counts. With our complete conditionals in place, we now derive the coordinate ascent algorithm for the expanded set of latent variables.

| Latent Variable | Type | Complete conditional | Variational parameters |
|---|---|---|---|
| $\theta_{dk}$ | Gamma | $c + \sum_v z_{dv,k} + \sum_u y^a_{ud,k}, d + \sum_v \beta_{vk} + \sum_u \eta_{uk}$ | $\tilde{\theta}^{\text{shp}}_{dk}, \tilde{\theta}^{\text{rte}}_{dk}$ |
| $\beta_{vk}$ | Gamma | $a + \sum_d z_{dv,k}, b + \sum_d \theta_{dk}$ | $\tilde{\beta}^{\text{shp}}_{vk}, \tilde{\beta}^{\text{rte}}_{vk}$ |
| $\eta_{uk}$ | Gamma | $e + \sum_d y^a_{ud,k} + \sum_d y^b_{ud,k}, f + \sum_d(\theta_{dk} + \epsilon_{dk})$ | $\tilde{\eta}^{\text{shp}}_{uk}, \tilde{\eta}^{\text{rte}}_{uk}$ |
| $\epsilon_{dk}$ | Gamma | $g + \sum_u y^b_{ud,k}, h + \sum_u \eta_{uk}$ | $\tilde{\epsilon}^{\text{shp}}_{dk}, \tilde{\epsilon}^{\text{rte}}_{dk}$ |
| $z_{dv}$ | Mult | $\log \theta_{dk} + \log \beta_{vk}$ | $\phi_{dv}$ |
| $y_{ud}$ | Mult | $\begin{cases} \log \eta_{uk} + \log \theta_{dk} & \text{if } k < K, \\ \log \eta_{uk} + \log \epsilon_{dk} & \text{if } K \le k < 2K \end{cases}$ | $\xi_{ud}$ |

Table 1: CTPF: latent variables, complete conditionals and variational parameters.

**Variational family.** We define the mean-field variational family $q(\beta, \theta, \eta, \epsilon, z, y)$ over the latent variables where we consider these variables to be independent and each governed by its own distribution,

$$q(\beta, \theta, \epsilon, \eta, z, y) = \prod_{v,k} q(\beta_{vk}) \prod_{d,k} q(\theta_{dk}) q(\epsilon_{dk}) \prod_{u,k} q(\eta_{uk}) \prod_{ud,k} q(y_{ud,k}) \prod_{dv,k} q(z_{dv,k}). \quad (2)$$

The variational factors for topic components $\beta_{vk}$, topic intensities $\theta_{dk}$, user preferences $\eta_{uk}$ are all Gamma distributions—the same as their conditional distributions—with freely set shape and rate variational parameters. For example, the variational distribution for the topic intensities $\theta_{dk}$ is Gamma$(\theta_{dk}; \tilde{\theta}^{\text{shp}}_{dk}, \tilde{\theta}^{\text{rte}}_{dk})$. We denote shape with the superscript "shp" and rate with the superscript "rte". The variational factor for $z_{dv}$ is a multinomial Mult$(w_{dv}, \phi_{dv})$ where the variational parameter $\phi_{dv}$ is a point on the $K$-simplex. The variational factor for $y_{ud} = (y^a_{ud}, y^b_{ud})$ is also a multinomial Mult$(r_{ud}, \xi_{ud})$ but here $\xi_{ud}$ is a point in the $2K$-simplex.

**Optimal coordinate updates.** In coordinate ascent we iteratively optimize each variational parameter while holding the others fixed. Under the conditionally conjugate augmented CTPF, we can optimize each coordinate in closed form by setting the variational parameter equal to the expected natural parameter (under $q$) of the complete conditional. For a given random variable, this expected conditional parameter is the expectation of a function of the other random variables and observations. (For details, see [9, 10]). We now describe two of these updates; the other updates are similarly derived.

The update for the variational shape and rate parameters of topic intensities $\theta_{dk}$ is

$$\tilde{\theta}_{dk} = \langle c + \sum_v w_{dv} \phi_{dv,k} + \sum_u r_{ud} \xi_{ud,k}, d + \sum_v \frac{\tilde{\beta}^{\text{shp}}_{vk}}{\tilde{\beta}^{\text{rte}}_{vk}} + \sum_u \frac{\tilde{\eta}^{\text{shp}}_{uk}}{\tilde{\eta}^{\text{rte}}_{uk}} \rangle. \quad (3)$$

The Gamma update in Equation 3 derives from the expected natural parameter (under q) of the complete conditional for $\theta_{dk}$ in Table 1. In the shape parameter for topic intensities for document $d$, we use that $\mathrm{E}_q[z_{dv,k}] = w_{dv}\phi_{dv,k}$ for the word indexed by $v$ and $\mathrm{E}_q[y^a_{ud,k}] = r_{ud}\xi_{ud,k}$ for the user indexed by $u$. In the rate parameter, we use that the expectation of a Gamma variable is the shape divided by the rate.

The update for the multinomial $\phi_{dv}$ is

$$\phi_{dv} \propto \exp\{\Psi(\tilde{\theta}^{\text{shp}}_{dk}) - \log \tilde{\theta}^{\text{rte}}_{dk} + \Psi(\tilde{\beta}^{\text{shp}}_{vk}) - \log \tilde{\beta}^{\text{rte}}_{vk}\}, \quad (4)$$

where $\Psi(\cdot)$ is the digamma function (the first derivative of the log $\Gamma$ function). This update comes from the expectation of the log of a Gamma variable, for example, $\mathrm{E}_q[\log \theta_{dk}] = \Psi(\tilde{\theta}^{\text{shp}}_{dk}) - \log \tilde{\theta}^{\text{rte}}_{dk}$.

**Coordinate ascent algorithm.** The CTPF coordinate ascent algorithm is illustrated in Figure 2. Similar to the algorithm of [9], our algorithm is efficient on sparse matrices. In steps 1 and 2, we need to only update variational multinomials for the non-zero word counts $w_{dv}$ and the non-zero ratings $r_{ud}$. In step 3, the sums over the expected $z_{dv,k}$ and the expected $y_{ud,k}$ need only to consider non-zero observations. This efficiency comes from the likelihood of the full matrix depending only on the non-zero observations [9].

Initialize the topics $\beta_{1:K}$ and topic intensities $\theta_{1:D}$ using LDA [2] as described in Section 3. Repeat until convergence:

1. For each word count $w_{dv} > 0$, set $\phi_{dv}$ to the expected conditional parameter of $z_{dv}$.

2. For each rating $r_{ud} > 0$, set $\xi_{ud}$ to the expected conditional parameter of $y_{ud}$.

3. For each document $d$ and each $k$, update the block of variational topic intensities $\tilde{\theta}_{dk}$ to their expected conditional parameters using Equation 3. Perform similar block updates for $\tilde{\beta}_{vk}$, $\tilde{\eta}_{uk}$ and $\tilde{\epsilon}_{dk}$, in sequence.

Figure 2: The CTPF coordinate ascent algorithm. The expected conditional parameters of the latent variables are computed from Table 1.

**Stochastic algorithm.** The CTPF coordinate ascent algorithm is efficient: it only iterates over the non-zero observations in the observed matrices. The algorithm computes approximate posteriors for datasets with ten million observations within hours (see Section 5). To fit to larger datasets, within hours, we develop an algorithm that subsamples a document and estimates variational parameters using stochastic variational inference [10]. The stochastic algorithm is also useful in settings where new items continually arrive in a stream. The CTPF SVI algorithm is described in the Appendix.

**Computational efficiency.** The SVI algorithm is more efficient than the batch algorithm. The batch algorithm has a per-iteration computational complexity of $O((W + R)K)$ where $R$ and $W$ are the total number of non-zero observations in the document-user and document-word matrices, respectively. For the SVI algorithm, this is $O((w_d + r_d)K)$ where $r_d$ is the number of users rating the sampled document $d$ and $w_d$ is the number of unique words in it. (We assume that a single document is sampled in each iteration.) In Figure 2, the sums involving the multinomial parameters can be tracked for efficient memory usage. The bound on memory usage is $O((D + V + U)K)$.

**Hyperparameters, initialization and stopping criteria:** Following [9], we fix each Gamma shape and rate hyperparameter at 0.3. We initialize the variational parameters for $\eta_{uk}$ and $\epsilon_{dk}$ to the prior on the corresponding latent variables and add small uniform noise. We initialize $\tilde{\beta}_{vk}$ and $\tilde{\theta}_{dk}$ using estimates of their normalized counterparts from LDA [2] fitted to the document-word matrix $w$. For the SVI algorithm described in the Appendix, we set learning rate parameters $\tau_0 = 1024, \kappa = 0.5$ and use a mini-batch size of 1024. In both algorithms, we declare convergence when the change in expected predictive likelihood is less than 0.001%.

## 4 Related work

Several research efforts propose joint models of item covariates and user activity. Singh and Gordon [19] present a framework for simultaneously factorizing related matrices, using generalized link functions and coupled latent spaces. Hong et al. [11] propose Co-factorization machines for modeling user activity on twitter with tweet features, including content. They study several design choices for sharing latent spaces. While CTPF is roughly an instance of these frameworks, we focus on the task of recommending articles to readers.

Agarwal and Chen [1] propose fLDA, a latent factor model which combines document features through their empirical LDA [2] topic intensities and other covariates, to predict user preferences. The coupling of matrix decomposition and topic modeling through shared latent variables is also considered in [18, 22]. Like fLDA, both papers tie latent spaces without corrective terms. Wang and Blei [20] have shown the importance of using corrective terms through the collaborative topic regression (CTR) model which uses a latent topic offset to adjust a document's topic proportions. CTR has been shown to outperform a variant of fLDA [20]. Our proposed model CTPF uses the CTR approach to sharing latent spaces.

CTR [20] combines topic modeling using LDA [2] with Gaussian matrix factorization for one-class collaborative filtering [12]. Like CTPF, the underlying MF algorithm has a per-iteration complexity that is linear in the number of non-zero observations. Unlike CTPF, CTR is not conditionally

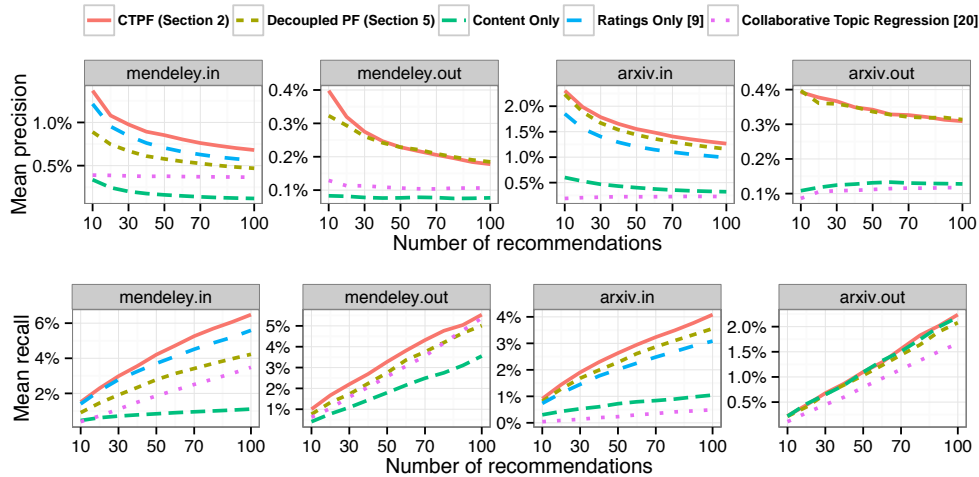

Figure 3: The CTPF coordinate ascent algorithm outperforms CTR and other competing algorithms on both in-matrix and out-matrix predictions. Each panel shows the in-matrix or out-matrix recommendation task on the Mendeley data set or the 1-year arXiv data set. Note that the Ratings-only model cannot make out-matrix predictions. The mean precision and mean recall are computed from a random sample of 10,000 users.

conjugate, and the inference algorithm depends on numerical optimization of topic intensities. Further, CTR requires setting confidence parameters that govern uncertainty around a class of observed ratings. As we show in Section 5, CTPF scales more easily and provides significantly better recommendations than CTR.

## 5  Empirical results

We use the predictive approach to evaluating model fitness [7], comparing the predictive accuracy of the CTPF coordinate ascent algorithm in Figure 2 to collaborative topic regression (CTR) [21]. We also compare to variants of CTPF to demonstrate that coupling the latent spaces using corrective terms is essential for good predictive performance, and that CTPF predicts significantly better than its variants and CTR. Finally, we explore large real-world data sets revealing the interaction patterns between readers and articles.

**Data sets.** We study the CTPF algorithm of Figure 2 on two data sets. The **Mendeley** data set [13] of scientific articles is a binary matrix of 80,000 users and 260,000 articles with 5 million observations. Each cell corresponds to the presence or absence of an article in a scientist's online library. The **arXiv** data set is a matrix of 120,297 users and 825,707 articles, with 43 million observations. Each observation indicates whether or not a user has consulted an article (or its abstract). This data was collected from the access logs of registered users on the http://arXiv.org paper repository. The articles and the usage data spans a timeline of 10 years (2003-2012). In our experiments on predictive performance, we use a subset of the data set, with 64,978 users 636,622 papers and 7.6 million clicks, which spans one year of usage data (2012). We treat the user clicks as implicit feedback and specifically as binary data. For each article in the above data sets, we remove stop words and use tf-idf to choose the top 10,000 distinct words (14,000 for arXiv) as the vocabulary. We implemented the batch and stochastic algorithms for CTPF in 4500 lines of C++ code.[1]

**Competing methods.** We study the predictive performance of the following models. With the exception of the Poisson factorization [9], which does not model content, the topics and topic intensities (or proportions) in all CTPF models are initialized using LDA [2], and fit using batch variational inference. We set $K = 100$ in all of our experiments.

- **CTPF**: CTPF is our proposed model (Section 2) with latent user preferences tied to a single vector $\eta_u$, and interpreted as affinity to latent topics $\beta$.

**Topic: "Statistical Inference Algorithms"**

**A) Articles about the topic; readers in the field**

| |
|---|
| On the ergodicity properties of adaptive MCMC algorithms |
| Particle filtering within adaptive Metropolis Hastings sampling |
| An Adaptive Sequential Monte Carlo Sampler |

**B) Articles outside the topic; readers in the field**

| |
|---|
| A comparative review of dimension reduction methods in ABC |
| Computational methods for Bayesian model choice |
| The Proof of Innocence |

**C) Articles about this field; readers outside the field**

| |
|---|
| Introduction to Monte Carlo Methods |
| An introduction to Monte Carlo simulation of statistical... |
| The No-U-Turn Sampler: Adaptively setting path lengths... |

**Topic: "Information Retrieval"**

**A) Articles about the topic; readers in the field**

| |
|---|
| The anatomy of a large-scale hypertextual Web search engine |
| Authoritative sources in a hyperlinked environment |
| A translation approach to portable ontology specifications |

**B) Articles outside the topic; readers in the field**

| |
|---|
| How to choose a good scientific problem. |
| Practical Guide to Support Vector Classification |
| Maximum likelihood from incomplete data via the EM… |

**C) Articles about this field; readers outside the field**

| |
|---|
| Data clustering: a review |
| Defrosting the digital library: bibliographic tools… |
| Top 10 algorithms in data mining |

Figure 4: The top articles by the expected weight $\theta_{dk}$ from a component discovered by our stochastic variational inference in the arXiv data set (Left) and Mendeley (Right). Using the expected topic proportions $\theta_{dk}$ and the expected topic offsets $\epsilon_{dk}$, we identified subclasses of articles: A) corresponds to the top articles by topic proportions in the field of "Statistical inference algorithms" for arXiv and "Ontologies and applications" for Mendeley; B) corresponds to the top articles with low topic proportions in this field, but a large $\theta_{dk} + \epsilon_{dk}$, demonstrating the outside interests of readers of that field (e.g., very popular papers often appear such as "The Proof of Innocence" which describes a rigorous way to "fight your traffic tickets"). C) corresponds to the top articles with high topic proportions in this field but that also draw significant interest from outside readers.

- **Decoupled Poisson Factorization**: This model is similar to CTPF but decouples the user latent preferences into distinct components $p_u$ and $q_u$, each of dimension $K$. We have,

$$w_{dv} \sim \text{Poisson}(\theta_d^T \beta_v); \quad r_{ud} \sim \text{Poisson}(p_u^T \theta_d + q_u^T \epsilon_d). \quad (5)$$

  The user preference parameters for content and ratings can vary freely. The $q_u$ are independent of topics and offer greater modeling flexibility, but they are less interpretable than the $\eta_u$ in CTPF. Decoupling the factorizations has been proposed by Porteous et al. [16].

- **Content Only**: We use the CTPF model without the document topic offsets $\epsilon_d$. This resembles the idea developed in [1] but using Poisson generating distributions.

- **Ratings Only [9]**: We use Poisson factorization to the observed ratings. This model can only make in-matrix predictions.

- **CTR [20]**: A full optimization of this model does not scale to the size of our data sets despite running for several days. Accordingly, we fix the topics and document topic proportions to their LDA values. This procedure is shown to perform almost as well as jointly optimizing the full model in [20]. We follow the authors' experimental settings. Specifically, for hyperparameter selection we started with the values of hyperparameters suggested by the authors and explored various values of the learning rate as well as the variance of the prior over the correction factor ($\lambda_v$ in [20]). Training convergence was assessed using the model's complete log-likelihood on the training observations. (CTR does not use a validation set.)

**Evaluation.** Prior to training models, we randomly select 20% of ratings and 1% of documents in each data set to be used as a held-out test set. Additionally, we set aside 1% of the training ratings as a validation set (20% for arXiv) and use it to determine convergence. We used the CTPF settings described in Section 3 across both data sets. During testing, we generate the top $M$ recommendations for each user as those items with the highest predictive score under each method. Figure 3 shows the mean precision and mean recall at varying number of recommendations for each method and data set. We see that CTPF outperforms CTR and the Ratings-only model on all data sets. CTPF outperforms the Decoupled PF model and the Content-only model on all data sets except on cold-start predictions on the arXiv data set, where it performs equally well. The Decoupled PF model lacks CTPF's interpretable latent space. The Content-only model performs poorly on most tasks; it lacks a corrective term on topics to account for user ratings. In Figure 4, we explored the Mendeley and the arXiv data sets using CTPF. We fit the Mendeley data set using the coordinate ascent algorithm, and the full arXiv data set using the stochastic algorithm from Section 3. Using the expected document topic intensities $\theta_{dk}$ and the expected document topic offsets $\epsilon_{dk}$, we identified interpretable topics and subclasses of articles that reveal the interaction patterns between readers and articles.

## Footnotes

[1]Our source code is available from: https://github.com/premgopalan/collabtm

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
