[Supplementary Material]



Figure 1: The collaborative topic Poisson factorization model (CTPF).

# A  Stochastic variational inference for the collaborative topic Poisson factorization model

Stochastic variational inference combines stochastic gradient algorithms and variational inference [3]. Stochastic gradient algorithms follow noisy estimates of the gradient with a decreasing step-size. If the expectation of the noisy gradient equals to the gradient and if the step-size decreases according to a certain schedule, then the algorithm converges to a local optimum [4]. To obtain noisy gradients, assume that we operate under the setting where we subsample a single document $d$ uniformly at random from the $D$ documents. This sampling strategy is similar to online LDA [2]. However, our approach differs in the use of separate learning rates for each user, allowing the inference to update only the relevant users in each iteration.

We are given observations about a single document in each iteration. Following [3], we use the conditional dependencies in our graphical model to divide our variational parameters into *local* and *global*. The multinomial parameters $(\phi_{dv}, \xi_{ud})$ for the sampled document $d$ and for all $u \in U$, and the Gamma parameters for $(\theta_{dk}, \epsilon_{dk})$ are local. All other varational parameters are global.

In each iteration of our algorithm, we first subsample a document. We then update the local multinomial parameters and the local topic intensities and offset parameters for this document using the coordinate updates from Figure 2. This optimizes the local parameters with respect to the subsample. We then compute scaled natural gradients [1] for the global user preference parameters $(\tilde{\eta}_{uk}^{\mathrm{shp}}, \tilde{\eta}_{uk}^{\mathrm{rte}})$ for the users $u$ that have rated document $d$ and for all topic parameters $(\tilde{\beta}_{vk}^{\mathrm{shp}}, \tilde{\beta}_{vk}^{\mathrm{rte}})$. The global step for the global parameters follows the noisy gradient with an appropriate step-size.

We maintain separate learning rates $\rho_u$ for each user, and only update the users who have rated the document $d$. We proceed similarly for words. We maintain a global learning rate $\rho'$ for the topic parameters, which are updated in each iteration. For each of these learning rates $\rho$, we require that $\sum_t \rho(t)^2 < \infty$ and $\sum_t \rho(t) = \infty$ for convergence to a local optimum [4]. We set $\rho(t) = (\tau_0 + t)^{-\kappa}$, where $\kappa \in (0.5, 1]$ is the learning rate and $\tau_0 \geq 0$ downweights early iterations [3].