[Reviews · NeurIPS 2014]

Submitted by Assigned_Reviewer_5

The authors propose a new topic model collaborative topic Poisson factorization (CTPF) which combines ideas from Poisson factorization (PF) and collaborative topic regression (CTR). The idea behind the generative model is intuitive and appealing: learn a latent representation for users and articles that explains the observed ratings as well as the content of the articles. The authors use stochastic variational inference which enables them to apply the model to large datasets. The authors claim that the use of a shared latent representation allows them to attack the cold-start recommendation problem. I like the idea a lot and I think it can potentially be very useful (e.g. as a web service for discovering new scientific articles) assuming the authors' claims are true. However, the presentation needs to be improved before I can whole-heartedly accept the paper.

- I felt that the paper is targeted at someone who knows the PF and CTR; the text in some sections reads like a "diff file" highlighting differences between the CTPF model from the older models :) It'd be better to present the CTPF model first and then perhaps describe how PF and CTR arise as special cases of CTPF.

- Figure 1: It might be better to move it to the discussion section. \theta and \epsilon are not defined in section 1. I was confused by what "topic offsets" meant until I read the latter sections. (Line 72 describes the red bars as "preferences of readers who have the EM paper in their libraries", which is not helpful either.)

- Evaluation: Do you assume that articles not rated are irrelevant? If so, does it not add noise to the computed precision and recall? Are these precision/recall values significant enough that users of such a system would notice an improvement in quality. Can you provide the performance of the "ideal" recommender which sorts by true relevance? Perhaps measures such as Normalized Discounted Cumulative Gain (NDCG) used in information retrieval would be more useful?

- The evaluation compares models just based on precision and recall. I think it would be better to compare other model properties as well (e.g. distribution over number of ratings per article, distribution of number of ratings per article)

- Line 88: Not clear to me why Poisson factorization handles sparse data better than Gaussian factorization. It's very common in recommender systems to let the objective function depend just on the observed ratings which requires us to update just the latent vectors for users/articles with non-zero observations.

- Line 130: PF model seems to use a separate rate parameter (gamma distribution) for each document and user. Is a separate rate not needed here?

Line 140: \epsilon is referred to as "document user weights" here and "topic offsets" elsewhere, which makes it hard to follow. It'd be better to use a consistent reference.

- How was K set in the experiments? How sensitive is the algorithm to Gamma hyper-parameters? Some sensitivity analysis on toy data might be helpful.

Line 374: "Content [1]" is misleading. The fLDA paper uses different priors, likelihood and a different inference scheme. I would encourage the authors to compare to fLDA if the source code is available. I guess "topic offset variables" refers to \epsilon?

- Line 414: why are train/test/validation proportions different for the two datasets? 1% of documents as a test set seems too small. Do you average results on multiple train/test splits?

- Figure 5: what do you mean by "readers in the field"? In A), B), C) it would be simpler if you specify what you mean in terms of \theta and \epsilon. There's some discrepancy between the figure and the caption ("ontologies and applications" vs "information retrieval").

- Line 431: The paper seems to end quite abruptly. It'd be good to have a discussion section to summarize the contributions and discuss future directions.

Typos, etc:

- Line 107: use r instead of y
- Line 285: \tau_0, \kappa not defined.

Update (after author rebuttal):

The authors have addressed most of my concerns. I have updated my rating, trusting the authors to make the necessary changes.

I think it'd be helpful to add a brief discussion early on about implicit feedback and how it differs from explicit feedback, for example the differences explained in Hu et al. [13]. IMO, the PF model is more appealing in the implicit feedback setting; the authors' response about why PF is better than competitors in the implicit feedback setting might be worth adding to the main text.

The fact that unobserved responses are treated as 0s in the evaluation is a bit unsatisfying and I am slightly worried how this assumption affects the authors' claims about 4x improvement over state-of-the-art. That said, I understand that evaluation on implicit feedback datasets is less straightforward. I strongly encourage the authors to make the data as well as the code publicly available so that others can study this problem in more detail.
Summary: The authors propose an interesting approach to improve recommender systems by using a topic model to model content. The authors combine tricks from several existing papers (recommendation module from PF, the idea of combining topic model with a matrix factorization model from CTR/fLDA, auxiliary variable trick + SVI for scalability) to develop (what appears to be) a novel approach to address the cold-start recommendation problem. Overall, the approach is promising, but the presentation needs to be improved.

Submitted by Assigned_Reviewer_23

Summary:
The authors present collaborative topic Poisson factorization (CTPF), a generative model of both text and user rating/log data. The running application in this paper is the important problem of recommending scientific articles to people based on previous rating/interaction data. CTPF draws mainly upon two recent models: collaborative topic regression (CTR) of Wang and Blei and Poisson factorization of Gopalan et al. Each document is represented by two latent vectors in K-dimensional topic space: \theta, based on the text of the document, and \epsilon, based on the document's readers. Each user is represented by a latent K-dimensional topic affinity vector, x. Observed word counts for each document are drawn from a Poisson centered on the product of theta and the topic-word matrix, while the observed user-document ratings are drawn from a Poisson centered on x * (\theta + \epsilon), leading to a very elegant combination of text data and readership data. Authors present both batch and stochastic variational inference algorithms for approximating the posterior, and then experimental results showing state-of-the-art recall and precision @20 performance on two real-world data sets.

--

Overall, the model is elegant and intuitive, the motivation is real and important, this work is executed well, the paper is written quite clearly, and the results are meaningful. I think this paper should be accepted. Some details below:

Quality:
- The model makes a lot of sense and elegantly addresses the ubiquitous cold start problem by additively combining factors based on the text of articles and the readership of articles in a Poisson factorization setting.
- Inference is efficient due to conditional conjugacy of Poisson factorization.
- Results show meaningful improvement over a state-of-the-art model like CTR (which incidentally won best paper at KDD 2011) and simpler/more factorized variants of the proposed model.
- Have you tried anything beyond binary ratings matrices? The Poisson model certainly doesn't assume binary ratings, and so would be interesting to see performance in that setting.

Clarity:
I can't overstate the importance of describing a model clearly and completely, and the authors do an excellent job of this here. A few points, though:
- line 81: what makes the topics from CTPF more interpretable than latent factors from matrix factorization?
- line 106-107: do you mean "greater than one"?
- line 169-170 and line 214: should say y_ud not y_dv. Also, the 1-2K indexing of y_ud mentioned on line 241 should probably be moved earlier, since the first time you see it is in Figure 2.
- would be nice to see a full listing of the variational updates in the appendix, to make it easy on others to implement the procedure

Originality and Significance:
- This paper builds upon two recent models, and in that sense is incremental, but it is executed well, and the combination of the two ideas (Poisson factorization and collaborative topic regression) leads to the most elegant joint model of text and user ratings I've seen.
Summary: Overall, the model is elegant and intuitive, the motivation is real and important, this work is executed well, the paper is written quite clearly, and the results are meaningful. I think this paper should be accepted. However, I'm curious how the model performs on non-binary rating data.

Submitted by Assigned_Reviewer_46

This paper proposes a new graphical model, that allows to uncover
latent topics for a text corpus, and latent interest from users. The
main novelty is to use jointly Poisson factorization in the document
and user spaces, by incorporating user/document offsets, thus allowing
to capture users' deviations from the pure topic intensities. The
standard variational tools from training this models are used, which
is quite easy in this case since the model is by construction
conditionally conjuguate.

The paper is very clean and nicely written, and the idea is nice and
quite simple. The idea, although seemingly based on several previous
tools existing from the huge graphical model literature, seem to be
new (although I not an expert of this field, so novelty shall be
assessed by another reviewer.)

I only have minor comments on the paper:

- p.~4 l-12 : a sum of Poisson is itself Poisson under independence,
which is the case since the variational distribution is fully
factorized. Please make this argument more clear

- It seems intuitively that initialization of the latent topics using
LDA helps a lot for obtaining good performance of the overall
algorithm in terms of accurary. A sentence explaining to what extent
this choice is important could be helpful, in particular, what
happends if no such initialization is done

- p.6 : I am not convinced by the explanation of the choice of
hyperparameters a, b, ... How do you choose these numerical values ?
Is there any cross-validation under the carpet for this ?

- p. l.7 l.10 in your text pre-processing, words are stemmed and
lemmatized ? Is there any hashing involved to reduce dimension ?

- p. l.7 I don't understand the sentence : "We treat user clicks
as..." Please explain more clearly how the user's ratings for
documents are constructed from this dataset. It can't be only
binary, as ratings are Poisson distributed in your model.
Summary: A good paper about a new graphical model for joint factorization with latent factors of document and user/document relations, with good numerical experiments
Author Feedback
Author rebuttal: We thank the reviewers for their constructive feedback.

> R23. Have you tried anything beyond binary ratings matrices?
> R46. It can't be only binary, as ratings are Poisson distributed in
> your model.

PF can capture marginal user, item distributions well and optimizing
the likelihood requires iteration over only the non-zero ratings
(Gopalan et al. (2013)). These advantages hold even in binary
settings; therefore, we prefer it over Gaussian/Bernoulli models.

We focussed on binary data sets of implicit feedback. Our extended
work will address non-binary data. Our ongoing work suggests that
"binarizing" provides better latent components and predictive
accuracy. In a pilot study, we developed a censored Poisson model. It
was computationally more expensive and it did not give better
empirical performance.

> R23. what makes the topics from CTPF more interpretable than
> latent factors from matrix factorization?

By tying latent factors to topics in content, CTPF allows factors to be
interpreted as distributions over words, rather than a list of document
titles. Further, the topics of new documents enable cold-start
recommendations.

> R5. Evaluation: Do you assume that articles not rated are
> irrelevant? If so, does it not add noise to the computed precision
> and recall?

We assume that the unrated articles are observed zeros. This includes
rated articles that are held out as part of the test set. The
precision/recall metrics measure the ability of the fitted model to
rank the rated test articles above the unrated articles.

> R5. Are these precision/recall values significant enough that users
> of such a system would notice an improvement in quality. Can you
> provide the performance of the "ideal" recommender which sorts by
> true relevance? Perhaps measures such as NDCG would be more useful?

Ideal recommender systems would have perfect (normalized) mean precision
(100%). Precision at 100 is a challenging task in domains with a large
number of items (260K and 600K in our case). Further we beat the state of
the art model with up to ~4x improvement in mean precision.

The NDCG measure may provide additional insight. We will include NDCG
scores in our revised paper.

> R5. I think it would be better to compare other model properties as
> well (e.g. distribution over number of ratings per article,
> distribution of number of ratings per article)

Thank you for this suggestion. We agree that model checking, for
example, using posterior predictive checks, is useful. We pursue this
approach in our extended work.

> R5. Line 88: Not clear to me why PF handles sparse data better than
> Gaussian factorization.

For sparse, implicit feedback data sets, Gaussian factorization
depends either on a careful assignment of confidence weights to each
observation (Hu et al. (2008)), or on randomly synthesizing negative
examples (e.g., Paquet et al.). PF naturally solves this problem by
capturing user activity with a Poisson distribution (recall that a sum of
Poissons is a Poisson). Further, under PF, updating the latent variables
with all matrix entries is almost as easy as updating with only non-zero
entries.

> R5. Line 130: Is a separate rate not needed here?

We presented the simplest content-based PF model. We can extend CTPF
with Gamma priors on the rate parameters, but this is not essential to
obtain gains from PF.

> R5. why are train/test/validation proportions different
> for the two datasets? 1% of documents as a test set seems too
> small. Do you average results on multiple train/test splits?

Only our validation sets differed across data sets. This was done to
account for the sparsity of the data. For the out-matrix experiments
on arXiv, 1% of the documents corresponds to 8600 articles; we found
it to be adequate for out-matrix experiments. We held out 20% of the
observed ratings for the in-matrix experiments.

Given the massive size of the data sets, we did not average results on
multiple train/test splits. Each data point in Figure 4 is an average
across 10,000 users.

> R5. Line 374: "Content [1]" is misleading. The fLDA paper uses
> different priors, likelihood and a different inference scheme. I
> would encourage the authors to compare to fLDA if the source code is
> available.

Thank you for this suggestion. We agree that fLDA is roughly similar
to a "Content-only" model, and will make this clear in our revised
version. Our understanding is that fLDA requires having user covariates
which we do not have. Similar to Wang et al. (2013), we compared to our
model with the topic offset variables fixed at zero.

> R46. ...what extent LDA initialization is important could be
> helpful, in particular, what happends if no such initialization is
> done

We agree; we will discuss the performance from random initialization
in the final version.

> R46, R5. How do you choose the Gamma hyperparameters ? How
> sensitive is the algorithm to the hyperparameter settings? How was K
> set in the experiments?

We fixed the Gamma hyperparameters following Gopalan et
al. (2014). The authors chose hyperparameters from a study on
Movielens data set varying scale/shape hyperparameters in the set
{0.1, 1, 10}. The algorithm is seen to be robust to hyperparameter
changes. We set K to 100 in all of our experiments.

On massive data sets, such exploration is infeasible. We plan to
develop an empirical Bayes based approach in our future work, but we
note the superior performance of PF models with fixed hyperparameter
settings.

> R46. p. l.7 l.10 in your text pre-processing, words are stemmed and
> lemmatized ? Is there any hashing involved to reduce dimension ?

We don't use stemming; we reduce the vocabulary size with tf-idf.

> R5. Figure 5: what do you mean by "readers in the field"?

We mean readers interested in the field, as in, those with a high
preference value at that topic.